DOI: 10.1038/s41467-018-04940-z　　**OPEN**

# Materials informatics for self-assembly of functionalized organic precursors on metal surfaces

Daniel M. Packwood[1,2] & Taro Hitosugi [3]

Bottom-up fabrication via on-surface molecular self-assembly is a way to create defect-free, low-dimensional nanomaterials. For bottom-up fabrication to succeed, precursor molecules which correctly assemble into the target structure must be first identified. Here we present an informatics technique which connects self-assembled structures with particular chemical properties of the precursor molecules. Application of this method produces a visual output (a dendrogram) that functions much like the periodic table, but whereas the periodic table puts atoms into categories according to the way in which they bond to each other, the dendrogram put molecules into categories according to the way in which they arrange in a self-assembled structure. By applying this method to the case of functionalized bianthracene precursors adsorbed to copper(111), we identify the functional groups needed to assemble one-dimensional chains, two-dimensional tilings, and other shapes. This methodology can therefore help to identify appropriate precursor molecules for forming target nanomaterials via bottom-up fabrication.

[1] Institute for Integrated Cell-Material Sciences (iCeMS), Kyoto University, Kyoto 606-8501, Japan. [2] Japan Science and Technology Agency (PRESTO), Kawaguchi, Saitama 332-0012, Japan. [3] School of Materials and Chemical Technology, Tokyo Institute of Technology, Tokyo 152-8352, Japan. Correspondence and requests for materials should be addressed to D.M.P. (email: dpackwood@icems.kyoto-u.ac.jp)

Bottom-up fabrication, which refers to the spontaneous formation of new materials via self-assembly of molecule precursors, is a way to create low-dimensional nanomaterials with atomic-scale structural precision[1,2]. In order to successfully assemble a specific nanomaterial via bottom-up fabrication, precursor molecules that interact and align correctly with each other during the self-assembly process must be first identified. It is therefore a major problem that the connection between precursor molecule structure and the outcome of the self-assembly process is yet to be fully elucidated.

An area where bottom-up fabrication is receiving a great deal of attention is the synthesis of graphene nanoribbons (GNRs)[3–9]. In one scheme for bottom-up GNR synthesis, bianthracene precursor molecules possessing bromine (Br) functional groups (10,10′-dibromo-9,9′-bianthracne, or Br$_2$BA) undergo a self-assembly process upon deposition onto a copper(111) (Cu(111)) surface, resulting in chain-shaped 'islands' (Fig. 1a). These chain-shaped islands consist of closely-packed rows of bianthracene molecules stabilized through $\pi$ stacking interactions and strong epitaxial interactions with the metal substrate. Upon heating, a chemical reaction occurs between the Br$_2$BA molecules, and the chain-shaped islands turn into GNRs with a (3,1)-chiral edge structure (The scheme shown in Fig. 1a differs from the one discussed in reference[3], which involves a gold surface, carbon–bromine bond cleavage, and covalent bond formation between precursor molecules instead of islands. Note that some authors have suggested that this bond cleavage scheme operates on copper(111) surfaces as well (see ref. [10]) [4–6,10]. The important point is that the islands apparently act as intermediate states during the GNR fabrication process[5]. Methods for controlling island shape should therefore be indispensable for bottom-up formation of GNRs with novel shapes.

A useful feature of the above system is that by simply varying the functional group attached to the bianthracene unit, we can systematically explore the connection between precursor molecule structure and the self-assembly outcome. In fact whereas bromine-functionalized bianthracene (Br$_2$BA) produces (3,1)-chiral edge GNRs when deposited on Cu(111), hydrogen-functionalized bianthracene (H$_2$BA) is known to produce the same GNRs but with slightly longer lengths[6]. The propensity to form chain-shaped islands therefore appears to be stronger for H$_2$BA than Br$_2$BA. On the other hand, methyl-functionalized bianthracene ((CH$_3$)$_2$BA) on Cu(111) does not produce chain-shaped islands at all, and tends to assemble into relatively formless islands[11]. Simulations using GAMMA (generalized block assembly machine learning equivalence class sampling) modeling[11] also predict a strong dependence of island shape on functional group (Fig. 1b). By connecting functional group properties with the outcome of bianthracene molecular self-assembly, we would be able to identify new bianthracene precursors for fabricating GNRs with novel shapes, and will also gain insights relevant to bottom-up fabrication on metal surfaces in general.

In this article, we use an informatics technique called hierarchical clustering to connect the chemical properties of precursor molecules with the outcome of the molecular self-assembly

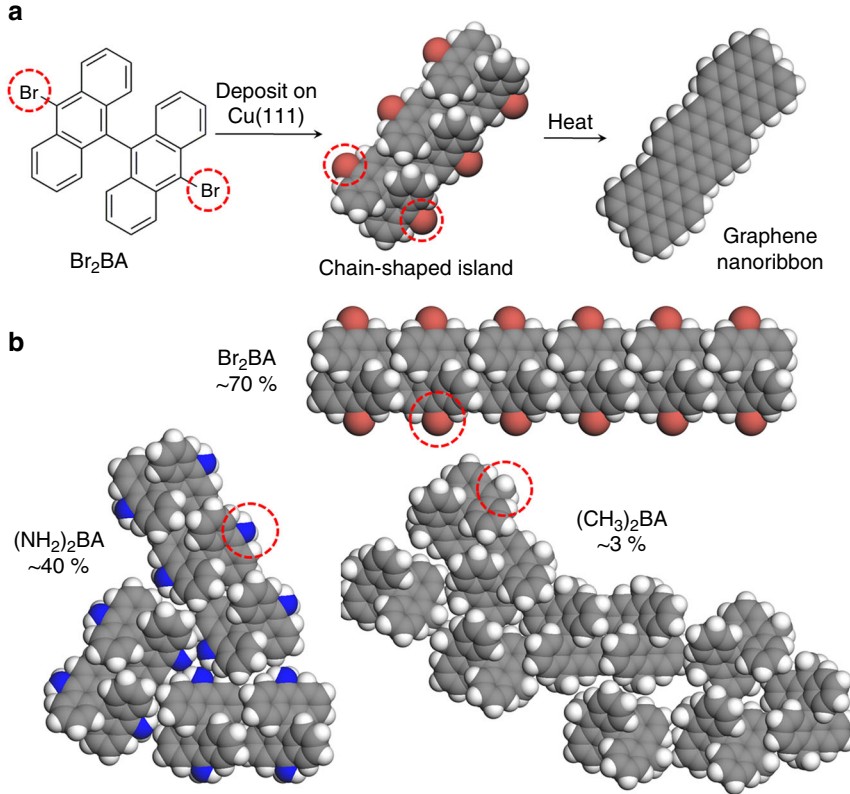

**Fig. 1** Self-assembly of bianthracene precursors on copper(111). **a** Bottom-up fabrication of graphene nanoribbons via deposition of dibromo-biranthracene (Br$_2$BA) on a copper 111 (Cu(111)) surface. The molecules form chain-shaped 'islands' via self-assembly, which undergo a chemical reaction to form graphene nanoribbons upon heating[5]. The Cu(111) surface is in the plane of the page, but not shown explicitly. Bromine functional groups are indicated by the red circles. **b** Some typical islands formed at 200 K by various bianthracene molecules possessing different functional groups (as predicted by the GAMMA model[11]). The percentages are probabilities of forming isolated chain-shaped islands. Functional groups are indicated by the red circles. Gray spheres = carbon, white spheres = hydrogen, red–brown spheres = bromine, and blue spheres = nitrogen atoms. All molecule structures were drawn in Materials Studio Visualizer[28]

process. Given a specific substrate and family of candidate precursor molecules, this analysis shows how to choose chemical properties from this family in order to assemble a desired type of structure. By 'family', we mean a series of molecules which are structurally homologous but differ only in chemical functionality. To demonstrate the application of our technique, we consider the substrate Cu(111) and the family of bianthracene precursor molecules ($X_2BA$). Members of this precursor family differ only in the type of functional group X attached at the 10 and 10′ carbons of the bianthracene unit. By application of our technique, we make the following deductions about functionalized bianthracene self-assembly on Cu(111) substrates: (i) electronically inert functional groups (such as H) are associated with the formation of chain-shaped islands in high yield; (ii) electronegative functional groups (such as F, Br, Cl, $CHCH_2$, and $NH_2$) are associated with a reduced yield of chain-shaped islands and the concurrent formation of chain clusters and defective chains; (iii) Functional groups whose orbitals can hybridize with the bianthracene orbitals (such as $CH_3$ or $CF_3$, via hybridization of their $\sigma$-bonds with the anthracene $p$ orbitals) are associated with formless island shapes, and almost never result in chain-shaped islands; (iv) Functional groups that form strong hydrogen bonds (such as CHO and OH) with H atoms of other molecules are associated with two-dimensional tilings. While the above deductions apply specifically to Cu(111) substrates and functionalized bianthracene precursor molecules, the technique itself can be used to make deductions for other types of substrates and precursor molecule families as well. Indeed, since functional group tuning is one of the most experimentally accessible ways of controlling molecular self-assembly processes, our technique is expected to find natural implementation in bottom-up nanomaterials research.

## Results

**Dissimilarity measure for hierarchical clustering.** Our technique uses data simulated via the GAMMA model[11,12]. The GAMMA model calculates the distribution of islands that appear at equilibrium following self-assembly of precursor molecules on a crystalline substrate held at a fixed temperature. Briefly, this

method considers $n$ identical molecules adsorbed to a metal surface of $N \times N$ unit cells and periodic boundaries (Fig. 2). The conformation of each molecule is identical, and the atoms of the molecules and the surface are static. The center of mass of each molecule may only reside over a finite number of points ('adsorption sites'), and the adsorbed molecules can only take on one of a finite number of orientations. The molecule conformation, adsorption sites and orientations are pre-determined via density functional theory (DFT) calculations. The energies of the model configurations are also calculated via DFT, where an interaction cut-off $M_c$ is utilized. An island is defined as a group molecules, such that each molecule in the group lies within the cut-off distance of at least one other molecule in the group. For convenience, a single isolated molecule is also regarded as an island. The distribution of islands is calculated by Monte Carlo sampling from the Boltzmann distribution at a fixed temperature $T$. At present, the GAMMA model is restricted to low coverages of molecule precursors. Further details are provided in the Methods section, as well as Supplementary Note 1 and Supplementary Figure 1.

In order to introduce our hierarchical clustering technique, suppose that we have a selected a specific surface (such as Cu(111)) and a specific family of precursor molecules (such as bianthracene molecules differing in the functional groups attached at the 10 and 10′ carbons) to study. Let X and Y denote two precursor molecules from this family. As the first step of hierarchical clustering, we quantify the dissimilarity of the islands formed by self-assembly of the X and Y precursor molecules. To this end, consider the 'island combination network' shown in Fig. 3. In the island combination network, the vertices (circles) $q_1$, $q_2$, … correspond to the unique island combinations that can be created from $n$ molecules, where $n$ is fixed. An edge (line) connects vertices $q_i$ and $q_j$ if and only if one of the following conditions is fulfilled: (1) $q_i$ can be transformed into $q_j$ by shifting a single molecule from one island to another, or (2) $q_i$ can be transformed into $q_j$ by removing a single molecule from an island to form a new 'island' containing only one molecule. Intuitively, the network in Fig. 3 describes how the various island combinations are related to each other by diffusion of molecules between islands. We define the dissimilarity $D(X, Y)$ between the islands formed by X and Y precursor molecules as

$$D(\mathrm{X}, \mathrm{Y}) = \sum_{q_i, q_j \in G} d\left(q_i, q_j\right) v_{\mathrm{X}}(q_i) v_{\mathrm{Y}}\left(q_j\right). \qquad (1)$$

The notation $q_i$, $q_j \in G$ means that the summation is performed over all pairs of vertices in the network (denoted $G$). $d(q_i, q_j)$ is the length of the shortest path in the network which connects vertices $q_i$ and $q_j$, and $v_{\mathrm{X}}(q_i)$ is the probability that the precursor molecules X will form island combination $q_i$ upon self-assembly on the surface at a fixed at temperature $T$ (the explicit calculation of $d(q_i, q_j)$ is demonstrated in Supplementary Figure 2). Explicitly, we have

$$v_{\mathrm{X}}(q_i) = C \exp\left(-\frac{F_{\mathrm{X}}(q_i)}{k_{\mathrm{B}} T}\right), \qquad (2)$$

where $C^{-1}$ is the partition function, $k_{\mathrm{B}}$ is the Boltzmann constant, $T$ is temperature, $F_{\mathrm{X}}(q_i)$ is the free energy of island combination $q_i$, defined as

$$F_{\mathrm{X}}(q_i) = \sum_{I \in q_i} \varepsilon_{\mathrm{X}}(q_i) - T k_{\mathrm{B}} \ln\, n_i, \qquad (3)$$

where $\varepsilon_{\mathrm{X}}(I)$ is the energy of island $I$, and $I \in q_i$ means that the summation is over all islands in $q_i$ and $n_i$ is the number of ways in

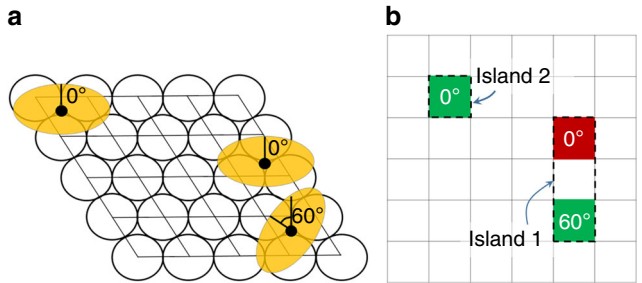

**Fig. 2** Illustration of the GAMMA model. **a** A configuration of molecules in the GAMMA model. The black-outlined circles are atoms of a (111) surface, the grid shows the unit cells of the surface, the orange ovals are adsorbed molecules, and the black points are the center of mass of the molecules. The orientation of each molecule is indicated by the numbers. A finite number of adsorption sites and orientations are available within each unit cell. **b** Shorthand notation for the model configuration in **a**. The grid corresponds to the unit cells of the surface, and the colored cells indicate unit cells holding an adsorbed molecule. The colors and numbers indicate the specific adsorption site within the unit cell and the orientation of the molecule, respectively. Groups of molecules that are close together (up to an interaction cut-off) distance are called islands. Isolated molecules are also regarded as islands. See main text for details

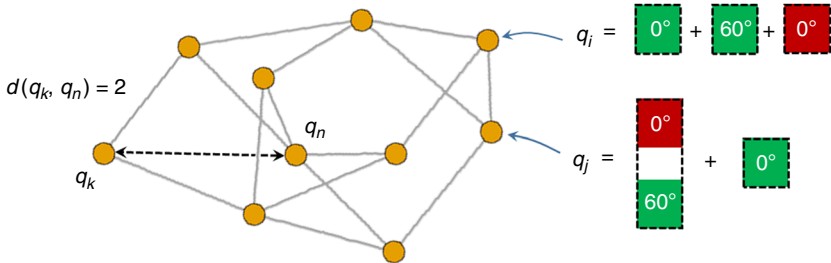

**Fig. 3** A simplified sketch of the island combination network. Vertices (circles) correspond to possible island combinations that can be formed from $n$ adsorbed molecules (in this case, $n = 3$), and two vertices are connected by an edge if and only if they differ by the state of a single molecule. The graph distance $d(q_i, q_j)$ is the length of the shortest path connecting vertices $q_i$ and $q_j$. The island combination network is used to construct the dendrogram shown in Fig. 4 (see main text for details)

which the islands in $q_i$ can be arranged on the surface. The chemical properties of precursor molecule X and the surface-molecule interaction are encoded in $\varepsilon_X(I)$. The value of $v_X(q_i)$ is output by the GAMMA model calculations. Note that $D(X, Y) = D(Y, X)$.

In order to make a physical interpretation of $D(X,Y)$ in Eq. (1), let us consider a system of $n$ precursor molecules of type X undergoing a self-assembly process on the surface. Suppose that we model the system trajectory as a discrete-time random walk on the island combination network, where this random walk evolves in such a way that the free energy (Eq. (3)) tends to decrease and the probability distribution of the walk converges to the Boltzmann distribution in Eq. (2). When the random walk crosses a single edge in the network, it means that a single molecule in the system changes its state. While this model does not accurately describe the short-time evolution of self-assembly process, it does possess the correct long-time (equilibrium) distribution (see Supplementary Note 2). Now, consider two independent random walks on the island combination network, corresponding systems of X and Y precursor molecules, respectively, and let $Q_k^X$ and $Q_k^Y$ represent the position of these random walks at time $k$, respectively. Then,

$$D^*(X, Y) = \lim_{t \to \infty} \frac{1}{t} \sum_{k=1}^{t} d(Q_k^X, Q_k^Y) \qquad (4)$$

measures the time-averaged separation of the two self-assembly processes in the island combination network (note that the effect of the short-time dynamics vanishes upon taking the limit in (4)). By applying the ergodic theorem, which essentially states that time averages across random walk trajectories are equivalent to ensemble averages, we find that $D^*(X,Y) = D(X,Y)$. In other words, $D(X, Y)$ in Eq. (1) measures the time-averaged separation of the X and Y system trajectories on the island combination network, during the molecular self-assembly processes.

In passing, note that the island combination network discussed above is extremely large and complex, and cannot be easily generated or stored in a computer's memory. This means that standard techniques for computing network distances (such as Djikstra's algorithm[13]) cannot be reasonably applied to the current situation. In the Supplementary Information, we derive an exact formula for $d(q_i, q_j)$ in terms of $q_i$ and $q_j$ alone (Supplementary Note 3, Supplementary Figures 3–4). With this formula, the network distances can be computed directly without having to generate the island combination network at all.

**Analysis of functionalized bianthracene precursors on copper.** In the following calculations, we consider the specific case of Cu (111) substrates and the family of functionalized bianthracene

precursor molecules (i.e., X$_2$BA, where X denotes the functional groups attached to the 10 and 10′ carbons of the bianthracene unit). In order to perform the hierarchical clustering, we calculate the dissimilarity $D(X, Y)$ of the islands formed by X$_2$BA and Y$_2$BA, for 10 different functional groups X and Y, and then plot the dissimilairites as a dendrogram (Fig. 4). The dendrogram can be understood by considering how it was created (Supplementary Note 4 and Supplementary Figure 5). In the first step, the two functional groups with the smallest dissimilarity (H and F) are identified and placed into a so-called cluster. This cluster is represented by the horizontal line connecting H and F together in Fig. 4, where the height of the horizontal line is equal to $D(H, F)$. In the second step, CHCH$_2$ and F are identified as having the second smallest dissimilarity, and so CHCH$_2$ is added to the cluster containing H and F. Again, this clustering is represented by the horizontal line connecting CHCH$_2$ with H and F, and the height of the horizontal line is given by $D(\text{CHCH}_2, F)$. On the third step, Br and NH$_2$ are identified as having the third smallest dissimilarity, and so Br and NH$_2$ are put into a single cluster. In the fourth step, Cl joins the cluster containing CHCH$_2$, H, and F, and in the fifth step, the cluster containing Br and NH$_2$ joins the cluster containing Cl, CHCH$_2$, H, and F. This process continues until all functional groups belong to the same cluster. The dendrogram therefore arranges the functional groups according to the similarity of the islands that are formed upon bianthracene self-assembly. The specific type of hierarchical clustering used here is called complete-linkage hierarchical clustering. Other hierarchical clustering methods did not result in major changes to the dendrogram (Supplementary Figures 6–8).

By comparing the dendrogram in Fig. 4 with the island shapes predicted by simulation (see Supplementary Table 1, Supplementary Figures 9–18), we find that the functional groups on the right-hand side of the dendrogram (H) mainly lead to chain-shaped islands whereas the functional groups on the left-hand side (CHO, HO) mainly lead to tiling patterns. Islands at these two extremes can be referred to as 1D and 2D crystals, respectively. Islands between these two extremes have intermediate shapes. For the case of functionalized bianthracene precursors on Cu(111), the effect of functional group tuning on island shape is therefore to induce of a phase transition from ordering in one dimension to ordering in two dimensions. Note that, as we proceed from right (the 1D crystal side) to left (2D crystal side) across the dendrogram, the height of the horizontal lines connecting the functional groups in the dendrogram tends to increase, giving the dendrogram a staircase-like appearance. In the hierarchical clustering literature, this 'staircase' effect is often taken as a symptom of poor clustering. However, the staircase effect is physically unavoidable in the present context. For the family of bianthracene precursors on Cu(111), we only ever see

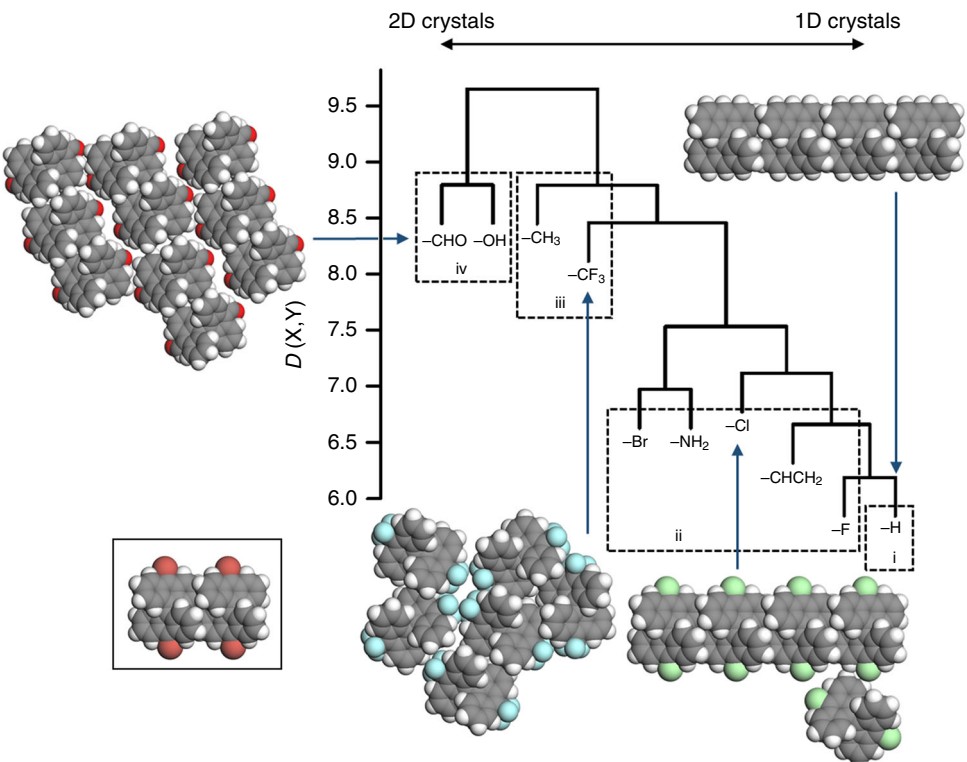

**Fig. 4** Hierarchical clustering for bianthracene molecules. In this diagram (dendrogram), bianthracene functional groups are arranged according to the similarity of the islands formed upon molecular self-assembly on Cu(111) at 200 K. Functional groups on the right-hand side have a strong tendency to form chain-shaped islands (1D crystals), whereas functional groups on the left-hand side have a strong tendency to form 2D tilings (2D crystals). Functional groups between these two extremes produce intermediate structures. In order to interpret the dendrogram, we divide the functional groups into four categories, as indicated by the dotted boxes. From right to left, the boxes correspond to the Strong 1D Crystal Formers (category i), Moderate 1D Crystal Formers (category ii), Weak 1D Crystal Formers (category iii), and Strong 2D Crystal Formers (category iv). Some island shapes for various functional groups are shown surrounding the dendrogram. The insert shows two bianthracene molecules aligned in the direction of their bianthracene tips. Gray spheres = carbon, white spheres = hydrogen, red–brown spheres = bromine, blue spheres = nitrogen, green spheres = chlorine, and bright red spheres = oxygen

one kind of 1D crystal (corresponding to a chain of molecules aligned in the direction of the bianthracene tips) in our simulations. This means that, if we try to induce a phase transition to 1D crystals by functional group tuning, we will always converge to the same, chain-like arrangement of molecules on the surface, regardless of the kinds of islands that we started with. This convergence is reflected by the staircase effect in the dendrogram.

To connect bianthracene functional group properties with island shapes, we categorize the functional groups by considering the order in which they are clustered in the dendrogram. Starting from the bottom-right of the dendrogram and working towards the top-left, we see that the functional groups are clustered in roughly four categories: H (category i), F, CHCH$_2$, NH$_2$, Br and Cl (category ii), then CH$_3$ and CF$_3$ (category iii), and finally OH and CHO (category iv). This categorization is arbitrary, however comparison with simulation results (Supplementary Table 1, Supplementary Figures 9–18) show that it adequately correlates with the types of islands formed by the functional groups. For example, category i functional groups clearly produce the 'most 1D / least 2D' islands, and category iv functional groups clearly produce the 'least 1D / most 2D' islands. We therefore refer to categories i, ii, iii, and iv as 'Strong 1D Crystal Formers', 'Moderate 1D Crystal Formers', 'Poor 1D Crystal Formers', and 'Strong 2D Crystal Formers', respectively. Note that this categorization applies to the specific case of Cu(111) surfaces and the family of functionalized bianthracene precursor

molecules, and may not hold for other types of substrates and precursor molecules.

By identifying the common chemical properties of the functional groups in each category, we can make the following deductions for functionalized bianthracene precursors on Cu (111) (Table 1).

The only functional group in the Strong 1D Crystal Formers identified in this study is H. An H atom in a C–H bond is electronically inert and does not facilitate any particular interactions between molecules. H$_2$BA molecules therefore have a strong preference to interact in the direction of their bianthracene tips (see the insert of Fig. 4), where $\pi$-stacking interactions are possible, rather than in any other direction. Simulations show that H$_2$BA self-assembly on Cu(111) at 200 K results in chain-shaped islands with around 96% probability (Supplementary Table 1, Supplementary Figure 9). While no other functional groups with such large chain-forming tendencies were found in this study, it seems that electronic inertness is necessary in order to achieve strong chain formation from functionalized bianthracene self-assembly on Cu(111).

The functional groups in the Moderate 1D Crystal Formers category (F, Br, Cl, CHCH$_2$, NH$_2$) all electronegative. In these cases, electronegativity allows for interactions (including hydro-gen bonding) between the functional group of one molecule and the atoms near the functional groups of another molecule (see Supplementary Figures 10–14). Compared to the Strong 1D Crystal Formers, these interactions lower the formation

**Table 1 Functional group properties associated with each category, and the types of islands formed upon self-assembly of bianthracene precursor molecules with functional groups with these properties (self-assembly on Cu(111) at low temperatures)**

| Category | Functional group properties for the bianthracene precursor | Types of islands resulting from self-assembly on Cu(111) |
|---|---|---|
| Strong 1D Crystal Formers (i) | Electronically inert (e.g., H) | Chains |
| Moderate 1D Crystal Formers (ii) | Electronegative (e.g., F, Br, Cl, $CHCH_2$, $NH_2$) | Chains (predominant) and clusters of chains or defective chains |
| Poor 1D Crystal Formers (iii) | Orbitals can hybridize with anthracene $\pi$-orbitals (e.g., $CH_3$, $CF_3$) | Formless clusters of short chains and molecules |
| Strong 2D Crystal Formers (iv) | Strong H-bonding with H atoms of other precursor molecules is possible (e.g., CHO, OH) | 2D tilings |

probability for chain-shaped islands by around 10–55% (Supplementary Table 1), because additional interactions (involving the functional groups) start to compete with the interactions directed along the bianthracene tips (Fig. 4 insert). However, these interactions are considered to be weak compared to interactions in the direction of the bianthracene tips (insert of Fig. 4), because chain-shaped islands are still strongly prevalent for functional groups in this category. Moreover, the 'non-chain-shaped' islands produced by these functional groups resemble clusters of chains or 'bent' chains (Supplementary Figures 10–14). In order to produce island shapes resembling clusters of 1D chains or bent chains from bianthracene self-assembly on Cu(111), it therefore appears important to use electronegative functional groups.

In contrast to the previous two categories, Poor 1D Crystal Formers ($CH_3$ and $CF_3$) overwhelmingly lead to disordered islands, and the formation of isolated chain-shaped islands is negligible (<3% formation probability on Cu(111) at 298 K). These large and disordered islands instead contain a compact and disordered mixture of chain-like segments mixed with additional molecules (Figs. 1, 4, Supplementary Figures 16–17). These island shapes are probably a result of hyperconjugation, i.e., hybridization of the C-H and C-F $\sigma$-bonds with the bianthracene $\pi$-orbitals. This would allow for electrons from the bianthracene $\pi$-orbitals to accumulate in the $\sigma$-bonds of these functional groups, reducing valence electron density on the bianthracene unit and increasing valance density on the functional group. In turn, this effect would stabilize interactions between the functional group of one molecule and the bianthracene tips of another molecule, inducing disorder in the island shape (Figs. 1, 4, Supplementary Figures 16–17). Such interactions appear to compete strongly with interactions directly along the bianthracene tips (Fig. 4 insert), resulting in a greatly reduced formation of chain-shaped islands. In order to form large and disordered islands from bianthracene self-assembly on Cu(111), functional groups with orbitals that can hybridize with the bianthracene $\pi$-orbitals appear necessary.

Finally, large two-dimensional molecular tilings form with high probability for functional groups contained in the 2D Crystal Former Category (CHO and OH). The islands in this category involve interactions in the direction of the bianthracene tips as well as interactions involving the functional groups. Moreover, these two interactions appear to be balanced in such a way that the islands resemble two-dimensional tilings. The common property of these functional groups is the ability to form hydrogen bonds. However, compared to the interactions involved with the electronegative functional groups in the Moderate 1D Crystal Former Category, the interactions involved in the present category appear much stronger. Note that the islands in the present category involve clear hydrogen bonds between the functional group O atom of one molecule and a non-functional-group H atom of a neighboring molecule (Supplementary Figures 17–18). By employing concepts from elementary chemistry, we find that for the case of the OH functional group,

the precursor molecules can be represented by a resonance structure in which the O atom has a positive partial charge and the C atom bonded to the H atom mentioned above has a partial negative charge (Supplementary Figure 17c). A similar situation is found for the case of the CHO functional group, however this time the O atom possesses a partial negative charge in the resonance structure, and the C atom bonded has a partial positive charge (Supplementary Figure 18c). These resonance structures favor the strong formation of hydrogen bonds between molecules in the islands. Note that similar resonance structures are possible for the case of $CHCH_2$ and $NH_2$ functional groups, however their influence on island shape should be relatively weak: in the case of $CHCH_2$, the resonance structure involves an energetically unstable terminal C atoms, and in case of $NH_2$ the resonance structure involves a relatively weak interaction between a partially positive N atom and a partially negative H atom (compared to the O atoms in the present case). In order to form 2D crystals from functionalized bianthracene self-assembly on Cu(111), it appears necessary to balance the strong interactions in the direction of the bianthracene tips with strong hydrogen bonds formed by the functional group.

While the importance of hydrogen bonding for forming 2D tilings in is well-known[1,2,14–18], the other connections between functional group properties and island shape elucidated above appear new and may be helpful for bottom-up fabrication of low-dimensional nanomaterials from functionalized bianthracene precursors on Cu(111). We stress that the dendrogram is only intended to connect island shapes with functional group properties, and is not meant to identify the physical mechanisms that lead to a particular island shape. While we used elementary chemistry concepts (such as electronegativity, hybridization, and resonance structures) to explain why particular functional group properties are associated with particular classifications of island shape, high-level quantum chemical calculations using localized basis sets are needed to rigorously validate these explanations. This is beyond the scope of the current paper.

The dendrogram is sensitive to surface temperature. This temperature dependence enters via the dissimilarity $D(X, Y)$ in Eq. (1), which is defined with-respect-to the Boltzmann distribution. The dendrogram shown in Fig. 4 was calculated at 200 K. Supplementary Figures 19–26 show dendrograms calculated at temperatures between 210 and 270 K. Note that dendrograms across a range of tempertures can be easily computed in our approach, because the GAMMA model simulations utilize the simulated annealing algorithm and therefore simultaneously sample the islands at a variety of temperatures[11]. The most pronounced change in the dendrograms with temperature is that, after 240 K, $CHCH_2$ and $CF_3$ are clustered together (Supplementary Figure 23), and shortly afterwards are clustered with $CH_3$. In order to form large, formless islands instead of chain-shaped islands, the energetic payoff of a large number weak pairwise interactions must outweigh the payoff of a small number of strong pairwise interactions.

However, as temperature increases, it becomes entropically favorable for a large number of small islands to be present on the surface[11,19]. Because these small islands possess few molecules, a large number of weak pairwise interactions cannot be achieved for these cases. Consequently, the small islands are stabilized by a small number of strong pairwise interactions, and therefore take on a chain shape. Small, chain-shaped islands therefore always dominate over large, formless islands at high temperature. Because of this effect, the CF$_3$ functional group fall into the Moderate 1D Crystal Formers category as temperature increases. A similar effect can be seen after 240 K, at which the Strong 1D Crystal Formers and the Moderate 1D Crystal Formers become very difficult to distinguish. For example, in the dendrograms calculated from 240 K, the functional groups H, F, Br, Cl, and NH$_2$ are clustered together in quick succession, and the order in which they are clustered changes as temperature increases further. This is again due to prevalence of smaller islands at higher temperatures, which have a tendency to take on a chain shapes. Because higher temperatures are associated with smaller island sizes and a tendency to form chain-shaped islands, we recommend applying the method presented here at low temperature, where the effect of functional group chemical properties on island shapes are most clear.

## Discussion

While the deductions made above strictly apply to Cu(111) substrates and functionalized bianthracene molecules, the technique itself may be applied in exactly the same way to other substrates and precursor molecule families as well. A demonstration of this method for the case of a Cu(100) substrate and functionalized bianthracene molecules is presented in the Supplementary Information (Supplementary Note 5, Supplementary Figures 27–32, Supplementary Table 2). For this case, we observe that H$_2$BA as well as Br$_2$BA both belong to the category of strong 1D crystal formers, which suggests that either electronically inert functional groups or electronegative functional groups may be used to form chain-shaped islands from bianthracene self-assembly on Cu(100). (CH$_3$)$_2$BA belongs to the category of weak 2D crystal formers, and forms islands with ordering in two-dimensions with higher probability than it does on Cu(111). This suggests that hyperconjugating functional groups may be used to form 2D crystals from bianthracene self-assembly on Cu(100). In general, our approach may be applied to any case which satisfies both of the following conditions: that the surface-precursor molecule interaction is much stronger than the precursor-precursor molecule interaction, and that the surface is perfectly crystalline. While this condition applies to a many of the systems used for nanomaterials fabrication at present (particularly those involving flat, aromatic precursor families and low-index metal surfaces), it excludes many cases involving semiconducting or insulating surfaces, in which the surface-molecule interaction is often weak. Our technique may therefore be used to deduce guidelines for controlling the molecular self-assembly process, providing that the surface and precursor molecule family in question satisfy the above conditions.

In this paper, we have illustrated our method for the case of functionalized bianthracene precursors on Cu(111). While this system is relevant to GNR fabrication (Fig. 1), a more common approach to GNR fabrication involves carbon biradical precursors, which form via surface-catalyzed dehalogenation reactions. In this case, radical recombination occurs following precursor self-assembly, resulting in covalently bonded polymers rather than supramolecular islands as the intermediate state in the fabrication scheme[3,10,20]. In principle, our framework is applicable to the case of radicalized precursors as well.

In this case, instead of considering a family of precursor molecules differing in functional group, we would consider a family of biradical precursors which differ in chemical structure. For example, we could consider a family consisting of naphthalene biradicals, anthracene biradicals, pentacene biradicals, and so on, and a gold(111) surface. For each of these biradicals, we would first use a simulation method to predict the structure and length distribution of the polymers which form upon self-assembly. From these predictions, we would compute a dissimilarity metric analogous to Eq. (1), from which we could compute a dendrogram analogous to Fig. 4. By interpretation of this dendrogram, we could then deduce rules which connect the chemical structure of the biradical precursor with the type of polymer formed, and in turn predict new radical precursors for forming GNRs with novel widths and edge structures. While such a dendrogram would be very useful for GNR engineering, there is an important caveat: at present, there is no standard theoretical methodology for simulating radical precursor self-assembly on metal surfaces. In this case, the intermolecular interaction makes the major contribution to the energetics of the system, and our GAMMA method for simulating self-assembly cannot be expected to perform well. In order to construct a dendrogram for families of radical precursors, we must first develop novel techniques for simulating molecular self-assembly processes for these systems.

In order to improve the usefulness of bottom-up fabrication in the creation of new nanomaterials, it is crucial that the outcome of the self-assembly process can be predicted from the chemical properties of the precursor molecule. In this paper, we have presented an informatics technique (hierarchical clustering) which can be used to connect precursor molecule properties with the shapes of islands formed by precursor molecule self-assembly. While we have applied our technique to the specific case of Cu (111) substrates and the functionalized bianthracene molecules, it may be applied in exactly the same way to other types of substrates and families of precursor molecules as well. In fact, for a given substrate and family of precursor molecules, the output of this technique (the dendrogram in Fig. 4) is analogous to the periodic table: whereas the periodic table groups the atoms according to the way in which they bond to each other, the dendrogram groups the molecules according to the way in which they are arranged in a supramolecular structure. Moreover, whereas the periodic table predicts the chemical properties needed for the constituent atoms to form a bulk material with a specific bonding structure, the dendrogram predicts the chemical properties needed for the precursor molecules to form a nanomaterial with a specific shape. However, in order to truly prove that that dendrograms or other informatics-based approaches can be as valuable to materials science as the periodic table, we must incorporate them in a real bottom-up nanomaterial fabrication experiment. We are currently pursuing this direction in our laboratories.

## Methods

**Computational model**. GAMMA model calculations used $n = 10$ molecules, $N = 50$ unit cells, and interatomic interaction cut-off of 8 Å, and $T = 200$ K. All relevant DFT calculations were performed in VASP using PAW-PBE pseudopotentials[21] and the rev-vdW-DF2 exchange-correlation functional[22–24]. The network distances ($d(q_i, q_j)$ in Eq. (1)) were calculated according to the shortcut formula in Supplementary Note 3. These calculations were performed in R using the external packages FNN and gtools[25–27].

**Data availability statement**. The source code and input files for performing simulations with the GAMMA model can be downloaded at http://www.packwood.icems.kyoto-u.ac.jp/download.

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

## Acknowledgements

This research was partially supported by the World Premier Research Institute Initiative promoted by the Ministry of Education, Culture, Sports, Science, and Technology of Japan (MEXT) for the Institute for Integrated Cell-Material Sciences (iCeMS), Kyoto University, and the Advanced Institute for Materials Research (AIMR), Tohoku University. Japan. D.P acknowledges Kakenhi No. 836167050004 and support from JST (PRESTO). T.H. acknowledges Kakenhi No. 26246022 and support from JST (CREST). The computation in this work was partially performed using the HA800-tc system RIIT at Kyushu University. M. Takahashi is thanked for additional computational support.

## Author contributions

D.P. and T.H. conceived the research topic together. D.P. created the theory and performed all calculations. D.P. drafted the paper. D.P. and T.H. wrote the final paper together.

## Additional information

**Competing interests:** The authors declare no competing interests.

