## [Peer Review File · Nature Communications]

Reviewer #1 (Remarks to the Author):

Report of referee

In this paper, Packwood and Hitosugi continue their efforts to develop technical tools for the prediction of 2D self-assembly patterns of organic molecules onto surfaces.

Now, they complement their previously developed GAMMA approach (Generalized block AsseMbly Machine learning equivalence clAss sampling) which generates equilibrium self-assembled patterns with a method to classify the obtained patterns.

The proposed method seems interesting, and the generated classification scheme (a dendrogram, see Figure 4) is clearly useful. The proposed scheme allows the identification of the effect of different functional groups on self-assembly, and how to obtain structures going from chain shaped structures to 2D crystals.

However, I have some concerns related to fundamental conceptual points as well as technical points that I think must be clarified by the authors before I can make a recommendation.

These are my points:

1- An essential point in the classification method is the definition of the dissimilarity D between structures defined in Eq.1. The definition may seem reasonable but it is also arbitrary to a large degree and requires more sound justification. The "physical interpretation" given to this definition in page 7 applies to almost any definition of D . Please, provide a more sound justification since the full procedure depends critically on D . Also, an illustration of the range of values of D will be useful. Looking at Figure 4, it is surprising to see that very different structures produce such a small difference in D . Going from strong 2D crystal formers (category iv) to weak 1D crystal formers involves a change in D from a value in the range 4-4.5 to 3-3.5. It seems that even the largest changes in structure involve only small changes in D (only 1 unit in D), which suggests that the classification scheme is not very accurate. At the end, the authors end up with a final classification with groups which is rather arbitrary and only weakly based on the values of D (see figure 4 for example). I think that the authors must demonstrate more convincingly that their definition of D is really able to distinguish clearly between different structures. This is critical to demonstrate the usefulness of the proposed approach.

2- In all the discussions, the chemical structure of the molecules plays an essential role but the influence of the surface is not obvious. I see that in their particular examples the surface plays only a minor role, but in many cases, self-assembly of molecules is only observed in certain surfaces or in particular reconstructions of the surfaces. The authors must demonstrate that their algorithm is also useful in this case.

3.-Temperature is also not discussed. It is not obvious to me whether the proposed scheme will be useful at other temperatures. At low temperatures, we will have different ordered structures ("crystals") whether at high temperature "disordered" structures will be common. How the proposed algorithm will work at different temperatures?

4.- A minor point is that the graph distance $d(q_1, q_2)$ between configurations q_1, q_2 plays an essential role in the proposed method, but it is not sufficiently explained in the main text. It is true that technical details are given in the supplementary info, but the concept is difficult to grasp with the present explanations. Additional, more concrete examples will help.

Reviewer #2 (Remarks to the Author):

The goal of the author's work is to apply an informatics strategy, hierarchical clustering, to categorize precursor molecules based on their functional groups for predicting the self-assembled structures they are likely to form. Such a strategy, if available, has the tremendous potential to accelerate materials discovery through theory-driven search strategies and, as such, is of high value. However, as is usually found in surface science and self-assembly, generally valid guiding principles are rare and successful general predictive strategies have thus far been elusive.

The research focus in this work is on bianthracene precursor molecules, which are known precursors for the fabrication of graphene nanoribbons. In fact, the development of synthesis strategies for GNRs "by design" currently a very active research area. The authors claim that based on the properties of functional groups of the precursor molecules they can predict the outcome of the self-assembly on Cu(111).

While I am convinced that this goal is valid and timely, and that the results presented are solid, the actual achievements presented are not yet the leap that is claimed.

1. While the authors point out that they discuss self-assembly of bianthracene on Cu(111), the influence of the substrate is not considered in the study. Substrates are known to massively influence molecular self-assembly, as has been shown in a number of publications on various organics generally, and on GNR-forming precursors specifically. The same type of precursors can give different results on Cu, Ag and Au surfaces, as can be seen on several examples in the literature.

2. One of the key conclusions, that electronically inert groups such as H lead to strong 1D crystal formation, is in contradiction with results in quite a few publications (not necessarily on GNR precursors). There is tons of evidence that molecules that are inert due to H termination pack closely, perhaps driven by van der Waals interaction.

3. The authors do not consider kinetic effects. For instance, the purpose of halogen functional groups (Br) is to detach on the surface, with the goal to allow for covalent bonding and 1D chain formation. As such, the result as supported by experiment should be that brominated bianthracene are strong 1D formers, where the chains are different from those in figure 1A of the manuscript. The molecules lose the Br on the surface and connect covalently on the former Br sites. Figure 1A is inconsistent with the experimental observations described in

J. Cai, P. Ruffieux, R. Jaafar, M. Bieri, T. Braun, S. Blankenburg, M. Muoth, A. P. Seitsonen, M. Saleh, X. Feng, K. Mullen, and R. Fasel, "Atomically precise bottom-up fabrication of graphene nanoribbons," *Nature*, vol. 466, no. 7305, pp. 470–473, Jul. 2010.

and

K. A. Simonov, N. A. Vinogradov, A. S. Vinogradov, A. V. Generalov, E. M. Zagrebina, N. Mårtensson, A. A. Cafolla, T. Carpy, J. P. Cunniffe, and A. B. Preobrajenski, "Effect of Substrate Chemistry on the Bottom-Up Fabrication of Graphene Nanoribbons: Combined Core-Level Spectroscopy and STM Study," *J Phys Chem C*, vol. 118, no. 23, pp. 12532–12540, Jun. 2014.

4. The authors conclude that "it is only necessary to identify the common chemical properties of the functional groups" to predict the structures formed, but this paper does not provide sufficient support for this hypothesis. As I mentioned, not all molecules with H-termination do the same thing. Not all molecules with CH₃ termination do the same thing, etc.

What makes the paper hard to read is that the authors "outsource" key results into the SI. While the main article is a concise summary, most of the results and evidence are put into the SI. On

page 13 for instance it is referred to figures S15-S22, to support their claim. This (and similar reference to other figures in the SI) makes it necessary to read the SI for the understanding of the article, or in other words, key figures to support any claims should be in the main article.

I conclude that it is too early for this study to hold what it promises and I recommend resubmission to another journal.

Response to Reviewer comments (NCOMMS-17-21811)

The reviewers are thanked for their constructive comments. On the basis of these comments, we have performed additional calculations and have made some major alterations to our manuscript. We believe that the new manuscript clearly communicates the theory and its broader significance, and is now appropriate for publication in *Nature Communications*. All new text is indicated in red.

Please note that the calculations for the (CHO)₂BA and Cl₂BA precursors have been repeated, as we noticed a typo in our original code. This results in minor changes to the dendrogram in Figure 4, but does not affect the rest of the paper at all.

1. Response to Reviewer # 1

1. An essential point in the classification method is the definition of the dissimilarity D between structures defined in Eq.1. The definition may seem reasonable but it is also arbitrary to a large degree and requires more sound justification. The “physical interpretation” given to this definition in page 7 applies to almost any definition of D . Please, provide a more sound justification since the full procedure depends critically on D .

The definition of D was actually motivated by the Monte Carlo sampling technique used in the GAMMA approach. Recall that Monte Carlo sampling involves simulating a random walk on the configuration space of the system. In the present case, this random walk starts at a point q_0 in the configuration space (or ‘island combination network’), and evolves in such a way that the free energy of the system $F(q)$ tends to decrease. After a sufficiently long simulation time, the probability that the random walk will be at point q_i in configuration space is given by $v(q_i) = C \exp(-F(q_i)/k_B T)$, where C is a normalizer constant, k_B is the Boltzmann constant, T is temperature. A key point is that this random walk can be regarded as an (approximate) model for the long-time dynamics of the self-assembly process. Moreover, the quantity $d(q_i, q_j)$ in equation 1 is equal to the minimum number of steps that this random walk must take in order to move from point q_i to point q_j in the configuration space (see the revised Supporting Information section S3).

Let us now consider the dissimilarity $D(X, Y)$ defined in equation 1, where X and Y denote two different types of precursor molecules. Suppose we run two independent random walks R_1 and R_2 on the configuration space, both starting at point q_0 . Furthermore,

suppose that R_1 and R_2 evolve according to the free energy functions for molecule X and Y, respectively, at some fixed temperature T . Then, after a long simulation time, the probability that the two random walks are separated by $d(q_i, q_j)$ steps is given by $v_X(q_i)v_Y(q_j)$. If we regard the random walks as models for the self-assembly process, $D(X, Y)$ then quantifies the time-averaged difference between the two self-assembly processes (note that the ensemble average in equation (1) can be interpreted as a time-average, due to the ergodic theorem).

The above shows that $D(X, Y)$ is a physically motivated way of comparing the self-assembly of precursor molecules of the type X and Y. We have incorporated this interpretation into the revised manuscript (pages 7 - 8), and have added some additional discussion in the Supporting Information (section S3).

As well as its strong physical motivation, our definition of $D(X, Y)$ has a practical justification as well: it is independent of the molecular structure of the precursor molecules. If a dissimilarity metric which is dependent on molecular structure (e.g., the spectral distance between island Coulomb matrices) were used, then the value of the metric would be a composite of both precursor molecule dissimilarity and island shape dissimilarity. Such a dissimilarity metric would therefore be unable to compare island shapes alone.

For the reasons outlined above, we believe that our definition of $D(X, Y)$ is appropriate for the analysis presented in our paper.

Also, an illustration of the range of values of D will be useful. Looking at Figure 4, it is surprising to see that very different structures produce such a small difference in D . Going from strong 2D crystal formers (category iv) to weak 1D crystal formers involves a change in D from a value in the range 4-4.5 to 3-3.5. It seems that even the largest changes in structure involve only small changes in D (only 1 unit in D), which suggests that the classification scheme is not very accurate. At the end, the authors end up with a final classification with groups which is rather arbitrary and only weakly based on the values of D (see figure 4 for example). I think that the authors must demonstrate more convincingly that their definition of D is really able to distinguish clearly between different structures. This is critical to demonstrate the usefulness of the proposed approach.

Firstly, we noticed a minor mistake in the dendrogram given in Figure 4 of the original manuscript, in which the vertical axis (D) values were multiplied by 0.5. This error does

not affect the discussion in the paper. We apologize for this error.

The reviewer remarks that D only changes by around 3 – 4 units across the whole dendrogram. While this is true, we emphasise that it is the absolute values of D which are most relevant to the interpretation of the dendrogram. For example, in the (corrected) dendrogram, we see that the 2D crystal formers (CHO and OH) are clustered with the rest of the functional groups at about $D = 9.5$. This means that the islands produced by CHO and OH differ from the islands produced by all other functional groups by *at most* 9.5 molecules. In other words, in order to transform the chain-shaped islands (produced by the H functional group) into a two-dimensional crystal (produced by the CHO functional group), at most over nine molecules in the islands must be transformed. This is quite large, considering that the calculations themselves only considered 10 molecules on the surface. This shows that large changes in island structure result in large changes in D , as we should have for an analysis of this type.

In order to improve comprehension of the dendrogram, we now specifically mention that the *complete linkage procedure* was used to perform the clustering (page 9). In the complete linkage procedure, the horizontal lines are drawn according to the maximum difference in the island shapes of the functional groups. Moreover, in the revised Supporting Information section S6, we present a histogram which was constructed using the *single linkage procedure* (which draws the horizontal lines according to the minimum difference between the island shapes) and the *average linkage procedure* (which draws the horizontal lines according to the average difference between the island shapes). These dendrograms are qualitatively similar to the dendrogram reported in Figure 4, and only shows minor differences in the positions of some functional groups and horizontal lines. More importantly, the single linkage dendrogram shows that the minimum differences between the various islands are also large, and again shows that our analysis is sufficiently sensitive to changes in island shape.

2. In all the discussions, the chemical structure of the molecules plays an essential role but the influence of the surface is not obvious. I see that in their particular examples the surface plays only a minor role, but in many cases, the self-assembly of molecules is only observed in certain surfaces or in particular reconstructions of the surface. The authors must demonstrate that their algorithm is also useful in this case.

Our method is applicable whenever the energetics of the system are dominated by the

surface-molecule interaction. In such a situation, molecules adsorb at well-defined positions on the surface, and the positions are not strongly affected by the presence of other molecules. This means that system can be treated with within the GAMMA model framework, and also that our concept of dissimilarity (which involves comparing molecule adsorption sites of different islands) in equation (1) is well-defined.

In order to demonstrate the applicability of our method to other types of surface reconstructions, we performed our analysis for the case of H₂BA, Br₂BA, and (CH₃)₂BA molecules adsorbed to Cu(100) (see the revised Supporting Information section S9). As with the Cu(111) cases described in the main text, the system energetics are again dominated by the surface-molecule interaction. The results obtained for the Cu(100) case are quite different than the Cu(111) case, and demonstrate the applicability of our method to different kinds of surfaces. For example, as well as H₂BA on Cu(100), Br₂BA on Cu(100) now forms chain-shaped islands with overwhelming probability (Table S2, Figure S29). (CH₃)₂BA is again found to be unable to form isolated chain-shaped islands on Cu(100), and therefore shows a large separation from Br₂BA and H₂BA in the dendrogram. However, in contrast to the case of a Cu(111) surface, (CH₃)₂BA on a Cu(100) surface forms islands which resemble clusters of chains and have long-range order in two dimensions. The new dendrogram in Figure S29 therefore suggests that on Cu(100) surfaces, Br₂BA and H₂BA can be classified as ‘Strong 1D Crystal Formers’, whereas (CH₃)₂BA can be classified as a ‘Weak 2D Crystal Former’,

Our method is therefore useful for other types of surface reconstructions, providing that the surface-molecule interaction dominates the system energetics. The latter is not a serious limitation, because such systems (typically flat aromatic molecules adsorbed to low-index metal surfaces) are widely used for bottom-up materials fabrication. As future work, we will aim to develop a new dissimilarity metric that can be applied when the surface-molecule and molecule-molecule interactions are comparable. This situation will be relevant for the technologically important case of semiconducting surfaces.

As a final comment, note that our method cannot be used to compare self-assembly processes taking place on different kinds of substrates. The dissimilarity metric D is only well-defined when the islands being compared are adsorbed on the same lattice. Our analysis therefore needs to be performed independently whenever a different kind of surface is considered. Again, we do not consider this to be a major disadvantage, because the main types of surfaces which are used in bottom-up nanomaterials synthesis (mainly

low index Cu, Au, or Ag) are few in number compared to the possible precursor molecules.

3.-Temperature is also not discussed. It is not obvious to me whether the proposed scheme will be useful at other temperatures. At low temperatures, we will have different ordered structures (“crystals”) whether at high temperature “disordered” structures will be common. How the proposed algorithm will work at different temperatures?

The effect of temperature on the dendrograms was discussed in the previous manuscript and the previous supporting information. In the new manuscript, these discussions can be found on pages 15 of the manuscript and Supporting Information Figures S19 – S26.

The current method can be straightforwardly applied at different temperatures. The temperature is implicit in the calculation of D in equation (1), which is defined as an average over a Boltzmann distribution at temperature T . Because the GAMMA approach implements parallel tempering (aka ‘simulated annealing’) to sample island shapes, our simulations actually generate several island samples across a range of temperatures. Dendrograms at various temperatures can therefore be easily computed from the output of the GAMMA simulations. We now emphasise this point on page 15 of the new manuscript.

4.- A minor point is that the graph distance $d(q_1, q_2)$ between configurations q_1, q_2 plays an essential role in the proposed method, but it is not sufficiently explained in the main text. It is true that technical details are given in the supplementary info, but the concept is difficult to grasp with the present explanations. Additional, more concrete examples will help.

As mentioned in our response to point 1 above, the graph distance $d(q_i, q_j)$ is the minimum number of steps that a random walk (representing the trajectory of the system on the configuration space) must take in order to move from point q_i to point q_j in the configuration space (see the revised Supporting Information section S3). In other words, it is the minimum number of molecules in q_i which must adjust their state in order for the system to transform into q_j .

In order to demonstrate the concrete meaning of $d(q_i, q_j)$, we have created added a new figure which shows the values of $d(q_i, q_j)$ for several pairs of islands (Supporting Information section S2 and Figure S2). This figure should facilitate the descriptions given

elsewhere in the manuscript.

2. Response to Reviewer # 2

While I am convinced that this goal is valid and timely, and that the results presented are solid, the actual achievements presented are not yet the leap that is claimed.

The Reviewer makes several suggestions that we have exaggerated the significance of our work. Consequently, we believe that there was a misunderstanding about the application of our analysis. In order to perform our analysis, we must first specify a *specific substrate and family of precursor molecules*. The predictions of the analysis then apply *to that specific substrate and family of precursor molecules*. It was not our intention to claim a set of universal rules for molecular self-assembly. Rather, we claim to have an analysis method from which rules for specific substrates and precursor families can be derived. This claim is theoretically rigorous and is demonstrated by calculations for bianthracene precursors on Cu(111). We have made several minor alterations to our manuscript to prevent such misunderstandings from occurring in the future (see pages 4, 6, 8, 9, 10, 11, and 15).

In order to be suitable for publication in *Nature Communications*, the work ‘should represent an advance in understanding likely to influence thinking in the field’. While the impact of the specific Cu(111)-bianthracene system might be debated, our analysis can be applied to any family of precursor molecules which differ in their chemical functional groups (providing that the surface-molecule interaction is sufficiently strong). Because functional group chemistry is arguably the most experimentally accessible means of controlling the molecular self-assembly process, we strongly believe that our method has the potential to influence future thinking in the area of bottom-up nanomaterials fabrication.

1. While the authors point out that they discuss selfassembly of bianthracene on Cu(111), the influence of the substrate is not considered in the study. Substrates are known to massively influence molecular selfassembly, as has been shown in a number of publications on various organics generally, and on GNR-forming precursors specifically. The same type of precursors can give different results on Cu, Ag and Au surfaces, as can be seen on several examples in the literature.

Please see our response to comment 2 of Reviewer 1. In the revised manuscript, we have demonstrated the applicability of our method to the Cu(100) substrate, which indeed shows very different results compared to the Cu(111) case (see the revised Supporting Information section S9). Our methodology can be applied to any other substrate in exactly the same manner, providing that the surface-molecule interaction is sufficiently strong.

2. One of the key conclusions, that electronically inert groups such as H lead to strong 1D crystal formation, is in contradiction with results in quite a few publications (not necessarily on GNR precursors). There is tons of evidence that molecules that are inert due to H termination pack closely, perhaps driven by van der Waals interaction.

The specific conclusion that we draw is that, *within the family of functionalized bianthracene precursors adsorbed to Cu(111)*, electronically inert groups lead to strong 1D crystal formation. This does not contradict the behavior reported for other families of precursor molecules.

The restriction of the analysis to specific families of molecules is not a major disadvantage; as mentioned above, bottom-up nanomaterials fabrication is usually performed on a relatively small number of metallic surfaces (mainly low-index Cu, Ag, and Au), and typically considers families of precursors which differ only in their chemical substituents. We now mention this point on pages 4, 6, 8, 9, 10, 11, and 15 of the revised manuscript.

Incidentally, there is indirect evidence in the literature that H₂BA (bianthracene) forms 1D crystals when adsorbed on Cu(111). Sanchez-Sanchez *et al* (*ACS Nano* **10**, 2016, 8006) studied GNRs formed according to the scheme shown in Figure 1, and found that their lengths tended to be longer than those formed by Br₂BA (see our response to point 3 below). This suggests a strong tendency for H₂BA to form 1D crystals on Cu(111).

3. The authors do not consider kinetic effects. For instance, the purpose of halogen functional groups (Br) is to detach on the surface, with the goal to allow for covalent bonding and 1D chain formation. As such, the result as supported by experiment should be that brominated bianthracene are strong 1D formers, where the chains are different from those in figure 1A of the manuscript. The molecules lose the Br on the surface and connect covalently on the former Br sites. Figure 1A is inconsistent with the experimental observations described in

J. Cai, P. Ruffieux, R. Jaafar, M. Bieri, T. Braun, S. Blankenburg, M. Muoth, A. P. Seitsonen, M. Saleh, X. Feng, K. Mullen, and R. Fasel, "Atomically precise bottom-up fabrication of graphene nanoribbons," Nature, vol. 466, no. 7305, pp. 470–473, Jul. 2010.

and

K. A. Simonov, N. A. Vinogradov, A. S. Vinogradov, A. V. Generalov, E. M. Zagrebina, N. Mårtensson, A. A. Cafolla, T. Carpy, J. P. Cunniffe, and A. B. Preobrajenski, "Effect of Substrate Chemistry on the Bottom-Up Fabrication of Graphene Nanoribbons: Combined Core-Level Spectroscopy and STM Study," J Phys Chem C, vol. 118, no. 23, pp. 12532–12540, Jun. 2014.

The Reviewer has exhumed an old controversy. Unfortunately, their comments contradict the prevailing view of GNR fabrication *via* bianthracene molecules on Cu(111).

While kinetic effects are certainly involved in Br₂BA self-assembly on Au(111) (as shown by Cai *et al* in the paper above), it is widely believed that *bromine atoms remain attached* to Br₂BA, even when it is adsorbed to Cu(111). Strong evidence supporting this, as well our Figure 1A, can be found in

Han *et al.* *ACS Nano* **8**, 2014, 9181

and

Han *et al.* *ACS Nano* **9**, 2015, 12035

In response to these papers, Simonov *et al* published a Comment in *ACS Nano* (volume 9, issue 4, pp 3399, 2015), which was effectively countered in a Reply by Han *et al* (pp 3404). In order to settle this issue, Sanchez-Sanchez and co-workers performed a definitive AFM study on this system, and unambiguously confirmed the GNR structure shown in Figure 1 (*ACS Nano* **10**, 2016, 8006). The GNR structure confirmed by Sanchez-Sanchez and shown in Figure 1 rules-out the radical recombination mechanism mentioned by the Reviewer. At present, Han's description of the Br₂BA / Cu(111) system prevail, and there is no convincing experimental evidence for the formation of bianthracene radicals from C-Br bond cleavage for this system.

Putting the experiment aside, we have never seen the Br atom detach from the Br₂BA molecule during our density functional theory structure relaxations on this system. Indeed, the precursor molecules shown in the paper are each in their energetically relaxed structure, and it can be seen that the Br atoms remain very much attached to the molecule. Even *ab initio* molecular dynamics simulations, which supply the C-Br atoms with energy in the order of $k_B T$, do not show evidence for Br atom detachment (Han *et al.* *ACS Nano* **8**, 2014, 9181). Computation therefore suggests that Br atom detachment at thermal energies is not kinetically favorable.

For the sake of readers outside of this field, the points described above are very briefly mentioned (see the note in reference [11] of the revised manuscript), and the Simonov *et al* paper mentioned above is now cited. In any case, the analysis method itself is the core of the paper, and can easily be applied to less controversial systems.

4. The authors conclude that “it is only necessary to identify the common chemical properties of the functional groups” to predict the structures formed, but this paper does not provide sufficient support for this hypothesis. As I mentioned, not all molecules with H-termination do the same thing. Not all molecules with CH₃ termination do the same thing, etc.

Please see our response to point 2 above. Our analysis is meant to help identify useful chemical properties *for a given substrate and family of precursor molecules*. We never intended to imply that all molecules with H-termination do the same thing, or that all molecules with CH₃-termination do the same thing.

We kindly ask for Reviewer 2 to evaluate our manuscript once again, while keeping the above points in mind.

Reviewer #1 (Remarks to the Author):

The authors have made a substantial effort to take into account my previous comments and the comments of the other referee. In general their responses are sound and the new details and discussions make the manuscript more clear and the computational procedure more transparent. I have still some (minor) concern about the reproducibility of the computational procedure. The computational procedure is essentially new and rather sophisticated (the details of the method were previously published in a recent paper by the same authors) and it is unlikely that anybody else has the possibility or expertise of repeating exactly the same steps (as it is in principle possible in more mature methods in which the simulation codes are freely available). Maybe the authors can consider adding some of their data/codes in repositories or as supporting information. Also, if the authors continue to use their method, I strongly suggest them to deliver the essential tools that they use (scripts, codes or example step-by-step tutorials) so other scientists may need to use their methods and reproduce their results. This is usually done in similar computational fields such as molecular dynamics simulations. As an example of best practices, I'll mention here the NAMD code for molecular simulation that has an extensive source of online free tools and free tutorials so anybody with access to computational clusters can in principle reproduce the methods and use them (there are of course many other examples of good practices). I have no further comments to the manuscript.

Reviewer #2 (Remarks to the Author):

The authors have addressed all my questions and concerns from my previous review very carefully, which is appreciated. Their responses, and the corresponding changes made to the manuscript, have helped answer my questions and improve the manuscript. Especially, the authors gave a convincing explanation as to why their manuscript is suitable for publication in Nature Communications, which was something I questioned in my initial review.

I am probably ready to recommend publication of their manuscript in Nature Communications. However, before I do so I am asking the authors to help me understand one more question, which I raised already in my first review:

My question (3) was about the role of the halogen atoms on the precursor molecules. The authors made a strong and convincing argument, based on results in the literature, regarding the structure the ribbons form and the role the halogen atoms play. Of course, the authors are right with their discussion of the ribbon formation of the bianthracene molecules.

However, my point was a slightly different one, and so I'm trying to rephrase here. As far as I interpret the results in the literature, importantly the results from the Fasel group (I am not from the Fasel group btw) and others, such as

Ruffieux, P., Wang, S., Yang, B., Sánchez-Sánchez, C., Liu, J., Dienel, T., et al. (2016). On-surface synthesis of graphene nanoribbons with zigzag edge topology. *Nature*, 531(7595), 489–492. <http://doi.org/10.1038/nature17151>

the purpose of the halogenes is to come off during annealing and thus to create defined attachment point for other precursor molecules. Precursor molecules connect where the halogenes once were. Obviously the bianthracenes doesn't seem to follow this scheme, but many other GNR precursors do. That's the key to current strategies to engineer GNRs, after all.

So let me ask my question differently: Do the authors see that their hierarchical clustering strategy would be able to predict the correct assembly if applied to another molecule/substrate system such as, say, those famous molecules 6,11-dibromo-1,2,3,4-tetraphenyltriphenylene in

Cai, J., Ruffieux, P., Jaafar, R., Bieri, M., Braun, T., Blankenburg, S., et al. (2010). Atomically precise bottom-up fabrication of graphene nanoribbons. *Nature*, 466(7305), 470–473.
<http://doi.org/10.1038/nature09211>

I have a hard time to see how their model would accommodate such a very different outcome for the same functional group, and I am hoping the authors can give some pointers (to me and ideally also in the paper) as to what it would take to adapt their model to similar GNR-forming systems and how their model would be able to predict the observed very different outcome of selfassembly for the halogene groups with other, related precursor molecules.

Reviewer #2 (Remarks to the Author):

All my questions have been fully addressed. I have no further questions and recommend publication of this manuscript.